



# Monitoring and quantifying CO₂ emissions of isolated power plants from space

Xiaojuan Lin[1,2], Ronald van der A[2,4], Jos de Laat[2], Henk Eskes[2], Frédéric Chevallier[3], Philippe Ciais[3], Zhu Deng[1], Yuanhao Geng[5], Xuanren Song[1], Xiliang Ni[6], Da Huo[1], Xinyu Dou[1], Zhu Liu[1,*]

[1]Department of Earth System Science, Ministry of Education Key Laboratory for Earth System Modeling, Institute for Global Change Studies, Tsinghua University, Beijing, 100084, China

[2]KNMI, Royal Netherlands Meteorological Institute, De Bilt, 3730 AE, the Netherlands

[3]Laboratoire des Sciences du Climat et de l'Environnement, CEA-CNRS-UVSQ, UMR8212, Gif-sur-Yvette, France

[4]Nanjing University of Information Science & Technology (NUIST), Nanjing, 210044, China

[5]Department of Statistics, School of Computer, Data & Information Sciences, University of Wisconsin - Madison, Wisconsin, 53706, USA

[6]Ministry of Education Key Laboratory of Ecology and Resource Use of the Mongolian Plateau & Inner Mongolia Key Laboratory of Grassland Ecology, School of Ecology and Environment, Inner Mongolia University, Hohhot, 010021, China

*Correspondence to*: Zhu Liu (zhuliu@tsinghua.edu.cn)

**Abstract**: Top-down $CO_2$ emission estimates based on satellite observations are potentially of great importance for independently verifying the accuracy of reported emissions and emission inventories. Difficulties in verifying these satellite-derived emissions arise from the fact that emission inventories often provide annual mean emissions while estimates from satellites are available only for a limited number of overpasses. Previous studies have derived $CO_2$ emissions for power plants from OCO-2 and OCO-3 observations of their exhaust plumes, but the accuracy and the factors affecting these emissions are uncertain. We have selected only isolated power plants for this study, to avoid complications link to multiple sources in close proximity. We first compare the Gaussian plume model and cross-sectional flux methods for estimating CO2 emission of power plants. Then we examined the sensitivity of the emission estimates to possible choices for the wind field. For verification we have used power plant emissions that are reported on an hourly basis by the Environmental Protection Agency (EPA) in the United States. By using the OCO-2 and OCO-3 observations over the past four years we identified emission signals of isolated power plants and arrived at a total of 50 collocated cases involving 22 power plants. We correct for the time difference between the moment of the emission and the satellite observation. We found the wind field halfway the height of planetary boundary layer (PBL) yielded the best results. We found that the instantaneous satellite estimated emissions of these 50 cases and reported emissions display a weak correlation ($R^2$=0.12). The correlation improves with averaging over multiple observations of the 22 power plants ($R^2$=0.40). The method was subsequently applied to 106 power plants cases worldwide yielded a total emission of 1522 ± 501 Mt $CO_2$/year, estimated to be about 17% of the power sector emissions of our selected countries. The improved correlation highlights the potential for future planned satellite missions with a greatly improved coverage to monitor a significant fraction of global power plant emissions.



**Keywords:** $CO_2$ emissions; Power plants; OCO-2; OCO-3; Gaussian plume model; $XCO_2$ enhancement

## 1. Introduction

The burning of fossil fuels for energy production has been driving the increase of atmospheric $CO_2$ concentrations from about 280 ppm to 410 ppm, which has dominated the observed planetary warming in the 20[th] and 21[st] centuries (IPCC, 2021). The Paris Agreement of the United Nations Framework Convention on Climate Change (UNFCCC) aims to keep global warming well within 2℃ above pre-industrial average temperatures by reducing global greenhouse gas (GHG) emissions. It has led to a strengthening of the reporting obligations of GHG emissions by the UNFCCC Parties (UNFCCC, 2018). These

reports are based on national fossil fuel $CO_2$ emission inventories that use, amongst others, input data about fossil fuel consumption, heating and carbon content of each fuel, type of combustion, and combustion efficiency. These reports are hampered by the difficulty to achieve accurate and detailed consumption data, especially for developing countries (Kort et al., 2012; Xue and Ren, 2012; Liu et al., 2015; Liu et al., 2020). The reports are self-declarations and although reviewed by expert teams within the UNFCCC process they lack independent verification. In order to address this issue, existing satellite retrievals

of the column-averaged dry air mole fraction of carbon dioxide $CO_2$ ($XCO_2$), mainly from NASA's Orbiting Carbon Observatory 2 & 3 (OCO-2 & OCO-3) and Japan's Greenhouse gases Observing SATellite (GOSAT), are increasingly explored for an independent verification of reported emissions (Nassar et al., 2017; Zheng et al., 2019; Shekhar et al., 2020; Kiel et al., 2021). Due to the promising first results of OCO-2 and GOSAT, new satellite instruments are being designed with a focus on a better sampling of the atmosphere (larger swath and / or constellation of satellites), providing an operational emission

monitoring capacity in the future (Engelen, 2021; CEOS, 2022; NASA, 2022).

Satellite observations are increasingly used for top-down estimates of fossil fuel emissions (Hakkarainen et al., 2021; Kuhlmann et al., 2021; Lauvaux et al., 2022). Zheng *et al.* (2020) revealed China's $CO_2$ emission drops and recoveries during the COVID 19 period using TROPOspheric Monitoring Instrument (TROPOMI) $NO_2$ data and bottom-up inventory data, and the GEOS-Chem model to compute the sensitivity of $NO_2$ concentrations to emissions. Other studies used satellite observations

of the total column dry-air $CO_2$ ($XCO_2$), a Bayesian inversion system and high-resolution transport model to quantify fossil fuel $CO_2$ (ff$CO_2$) emissions in urban areas (Kunik et al., 2019; Shekhar et al., 2020; Ye et al., 2020). For point source emitters, Bovensmann *et al.* (2010) introduced a conceptual technique to quantify $CO_2$ emissions of single power plants from $XCO_2$ plume enhancements. Janardanan *et al.* (2016) later connected $XCO_2$ enhancements observed by GOSAT with large emission sources through an atmospheric transport model in forward mode. Nassar *et al.* (2017, 2021) extended the approach and applied

it in backward mode in order to quantify $CO_2$ emissions from individual power plants using OCO-2 $XCO_2$ data. Reuter et al. (2019) used a few co-located regional enhancements of $XCO_2$ and $NO_2$ observed by OCO-2 and TROPOMI to estimate the $CO_2$ cross-sectional flux of the plumes. Zheng *et al.* (2020) used five year-worth of OCO-2 $XCO_2$ data to estimate emissions



from urban and industrial areas in China, and compared these local emissions with MEIC (Multi-resolution Emission Inventory for China; Zheng et al., 2018), EDGAR (Emissions Database for Global Atmospheric Research; Crippa et al., 2020) and

ODIAC (Open-Data Inventory for Anthropogenic Carbon dioxide; Oda et al., 2018) inventory estimations. The approach was extended to the entire globe and to OCO-3 by Chevallier *et al.* (2020, 2022).

Satellite instruments that directly detect $CO_2$ concentrations, such as OCO-2 and OCO-3, can be used for $CO_2$ emission estimates for power plants and other point sources. However, the limited track width, low revisit rate, and clouds mean these instruments only incidentally provide useful snapshot observations over a point source (see section 2.1 for details). Even fewer

observations are located in the downwind direction of the point source (in the plume), which are optimal to estimate emissions (Schwandner et al., 2017; Reuter et al., 2019; Zheng et al., 2019). Hence the demonstration of the use of satellite-observed $XCO_2$ to quantify point source $CO_2$ emissions comes from only limited cases or from Observing System Simulation Experiments (OSSEs; Bovensmann et al., 2010; O'Brien et al., 2016; Broquet et al., 2018; Kuhlmann et al., 2019; Wang et al., 2020; Wu et al., 2020). For example, Nassar *et al.* (2017, 2021) hand-picked a few OCO-2 tracks that captured the emission

plume of large coal plants for quantitative analysis. Other studies tried to be more systematic and iterated through the multi-year $XCO_2$ data and detected some of the space-time variations of anthropogenic emissions from locally-aggregated signals of emitters (Chevallier et al., 2020, 2022; Zheng et al., 2020), but it is difficult to attribute these signals to specific emission sources. In addition, instantaneous emissions at satellite overpass times are difficult to compare with an inventory of annual emissions, because of the intermittence and variability of power production and $CO_2$ emissions. This is why previous studies

had to use either instantaneous emission reports or temporally disaggregated inventories.

In this study, we advance the research of monitoring and quantifying point source carbon emissions by focusing on how to improve the accuracy of carbon emission using different wind data estimates, assess these emission estimates by comparing with USA EPA emission data, and identify and explore suitable cases elsewhere in the world. We compare the Gaussian plume model method with the cross-sectional flux method to estimate emission of power plants, in order to select the best method.

We analyze the impact of different wind field choices on the emission accuracy by comparing with hourly-reported emissions of selected power plants. Using the selected method we extend the estimation of power plant emissions to the global scale.

This paper is organized as follows. Section 2 describes data sources for this study. In section 3, we describe $XCO_2$ enhancement extraction, quantification and validation method as well as uncertainty calculation. The estimated $CO_2$ emissions for US and global power plants are presented in Section 4. Section 5 gives summary and conclusions.

**2. Data**

**2.1. Satellite data**

The Orbiting Carbon Observatory–2 (OCO-2) launched in July 2014 collects high-resolution spectra of reflected sunlight



in the bands centered near 0.765 μm, 1.61 μm, and 2.06 μm (Crisp et al., 2017). OCO-2 flies on a near-polar sun-synchronous

orbit and crosses the equator at a fixed local time (LT) near 13:36 with a repeat cycle of 16 days. About 10% of the

approximately 1 million daily $CO_2$ observations are cloudless and can be used to retrieve the column-averaged dry air mole

fraction of carbon dioxide ($XCO_2$) with the Atmospheric $CO_2$ Observations from Space (ACOS) algorithm at a spatial

resolution of $1.29 \times 2.25$ km$^2$ across swaths which are up to 10-km wide (O'Dell et al., 2018). The Orbiting Carbon

Observatory-3 (OCO-3) launched in May 2019 is mounted on the Japanese Experimental Module Exposed Facility (JEM-EF)

of the International Space Station (ISS) and views the Earth at all latitudes less than 52 degrees with a footprint size of $1.6 \times$

2.2 km$^2$. In addition to the same three observation modes (nadir, glint and target) as OCO-2, OCO-3 also collects nearly

adjacent swaths of data using a new pointing mirror assembly (PMA), resulting in a snapshot area map (SAM) scan of

approximately $80 \times 80$ km$^2$. Since all the power plants in this study are located on land, we only exploit OCO-2 and OCO-3

measurements over land surfaces (*i.e.* surface type =1 in the OCO data). For the observations of OCO-3 SAM mode, we only

analyse data on the same scan line (*i.e.* similar PMA elevation angles). We use good quality retrievals    (xco2_quality_flag =

0) of version 10r of the OCO-2 bias-corrected $XCO_2$ retrievals from January 2018 to December 2021 and version 10.4r of the

OCO-3 bias-corrected $XCO_2$ retrievals from August 2019 to November 2021 provided by the NASA Goddard Earth Sciences

Data            and            Information            Services            Center

(https://oco2.gesdisc.eosdis.nasa.gov/data/s4pa/OCO2_DATA/OCO2_L2_Lite_FP.10r/,

https://oco2.gesdisc.eosdis.nasa.gov/data/s4pa/OCO3_DATA/OCO3_L2_Lite_FP.10.4r/ ).

The passive-sensing hyperspectral nadir-viewing instrument TROPOMI on the Copernicus Sentinel-5 Precursor satellite

provides daily global coverage of tropospheric $NO_2$ vertical column densities ($NO_2$ TVCDs) with a spatial resolution of $3.5 \times$

7 km$^2$ initially and $3.5 \times 5.5$ km$^2$ since 6 August 2019. TROPOMI flies on a sun-synchronous orbit with an overpass time of

13:30 LT, same as OCO-2. Global daily $NO_2$ TVCDs maps were gridded to a regular longitude-latitude grid with 0.025°

resolution using pixels with a cloud fraction less than 30%. In order to keep as many pixels as possible to observe as much of

115    the plume as possible, data quality (*i.e.* "qa value") filtering is not used. The $NO_2$ data we obtained is consistent with the OCO-

2 and OCO-3 data of the same day in this study (https://s5phub.copernicus.eu/).

### 2.2. Power plant database

We collected reported hourly $CO_2$ emission data for U.S. power plants from the U.S. Environmental Protection Agency

(EPA) from 2018 to 2021 (https://ghgdata.epa.gov/ghgp/main.do#/listFacility/) as truth to validate the satellite estimated

emission. We sorted out the list from EPA for all 1631 power plants operated during this period.

The publicly available Global Power Plant Database (GPPD, v 1.3.0) from the World Resources Institute was used to

automatically identify $CO_2$ emission signals from power plants upwind of satellite tracks (Yin et al., 2021). The GPPD includes

34936 power plants (https://datasets.wri.org/dataset/globalpowerplantdatabase) with 15 fuel types, such as biomass,



geothermal, hydro, nuclear and solar. In this study, we selected power plants using gas, coal, oil and pet coke as primary fuel

to a subset. The resulting subset of GPPD comprises 8660 power plants of which 3998, 2330, 2320 and 12 use gas, coal, oil

and pet coke as primary fuel, respectively. In order to see emission signals from as many power plants as possible, we did not

delete power plants with a low generation capacity as done in other studies (Nassar et al., 2017; Beirle et al., 2021). The reason

is that if a power plant with a small capacity is located in an area with less background interference - such as an area far away

from cities and with low vegetation coverage - its emissions may still be detectable by satellites.

**2.3.  Wind fields**

We used three types of wind field data in the $CO_2$ emission estimation procedures described in Sect. 3. Hourly horizontal

wind fields $u$ and $v$ at 10 m are taken from the European Centre for Medium-Range Weather Forecasts (ECMWF) next-

generation reanalysis ERA5 dataset ($0.25° \times 0.25°$) and the Modern-Era Retrospective analysis for Research and Applications

version 2 MERRA-2 dataset ($0.5° \times 0.625°$). The effective wind speed proposed by Varon *et al.*(2018) and used by Reuter *et*

*al.* (2019) and Hakkarainen *et al.* (2021) was calculated from the 10 m wind by applying the empirical scaling factor 1.4. Varon

*et al.*(2018) derived this scaling factor from a linear fit between effective wind speed and 10 m wind speed. The effective wind

speed from ERA5 (WERA) and MERRA2 (WMERRA) are derived using factor 1.4 for this study. In addition, the wind field

at half the height of the PBL (WPBL) is derived from the 3D hourly wind fields and PBL heights derived from the twice daily

0-12 hour operational high-resolution forecasts of ECMWF. We use the wind vector of the model layer which contains the

altitude equal to the PBL height divide by 2. These three wind field choices are used to explore the robustness of the emission

estimates.

**3.  Method**

**3.1. Extract $CO_2$ enhancement from isolated power plant point sources**

The first step towards estimating $CO_2$ emissions from power plants is to extract plume $XCO_2$ anomalies, *i.e.* $XCO_2$ local

enhancement. We use power plants with hourly reported emissions from the USA EPA as research objects, and search for all

adjacent $XCO_2$ observations as candidate cases, and choose $XCO_2$ enhancement cases linked to isolated power plant emission

plumes by the wind direction. The search range is limited to a 0.25˚ radius around each power plant. The selection is based on

the visual identification of a plume in the downwind direction of the power plant. Power plants located in urban areas are

excluded because their emission plumes are compounded by urban emissions. This selection of cases for isolated U.S. power

plants allows us to verify the accuracy of estimated emissions using hourly reported emission data from EPA. For global $XCO_2$

data, we provide a semi-automatic detection algorithm described in Sect. 3.3.

**3.2. Gaussian plume model and Cross-sectional flux method**



(1) Gaussian plume model method

We use a Gaussian plume model and cross-sectional flux method for inferring power plant $CO_2$ emissions from $XCO_2$ measurements. In the Gaussian plume model (GPM) method (Bovensmann et al., 2010), the *posteriori* emission is obtained by a linear least squares fit between observed and simulated enhancements weighted by the reciprocal of the $XCO_2$ uncertainty. The model is based on the following equations:

$$V(x,y) = \frac{F}{\sqrt{2\pi} \cdot \beta \cdot (\frac{x}{1000})^{0.894} \cdot u} e^{-\frac{1}{2}\left(\frac{y}{\beta \cdot (x/1000)^{0.894}}\right)^2} \tag{1}$$

$$XCO_2 = V \cdot \frac{m_{air}}{m_{CO_2}} \cdot \frac{g}{P_{surf} - \omega \cdot g} \cdot 1000 \ , \tag{2}$$

where $V$ is the $CO_2$ vertical column at the location (x, y) downwind of the power plant (g m$^{-2}$), $x$ and $y$ are the along-wind distance and across-wind distance (m), respectively. $F$ is the emission rate (g s$^{-1}$), $\beta$ is the atmospheric stability parameter depending on Pasquill stability classes, which can be determined from the 10 meter wind speed and solar radiation obtained from ERA5 reanalysis data (Pasquill, 1961; Nassar et al., 2021) and $u$ is wind speed (m s$^{-1}$). Formula (2) is used to convert $V$ in g m$^{-2}$ to $XCO_2$ in ppm, in which $g$ is the gravitational acceleration (m s$^{-2}$), $m$ is the molecular weight (kg mol$^{-1}$), $P_{surf}$ is the surface pressure (Pa), and $\omega$ is the total column water vapor (kg m$^{-2}$) obtained from $XCO_2$ data files.

The wind direction is allowed to rotate within a range of $\pm 60°$ to account for errors in the wind data. The optimal wind direction is derived by maximizing the correlation coefficient between the simulated and the observed $XCO_2$ enhancement. We rejected the case if the maximum correlation coefficient is less than 0.5, similar as in Nassar et al. (2021). The outline and direction of the plume can be clearly seen in $NO_2$ images (Figure S8, S9), showing that the optimal wind direction is reliable in those cases.

(2) Cross-sectional flux method

In the cross-sectional flux method, the emission is inferred by integrating the plume enhancement over the background. An interval of 200 km along the track, centered on the maximum $XCO_2$ point, is taken as the analysis window. The following function is fitted to $XCO_2$ data in the analysis window (Figure S1a):

$$f(l) = k \cdot l + b + \frac{A}{\sigma\sqrt{2\pi}} e^{[-(l)^2/2\sigma^2]} \ , \tag{3}$$

where f($l$) is parameterized representation of $XCO_2$ (ppm); $l$ is the distance along the OCO-2 or OCO-3 tracks (km); $k, b, A,$ and $\sigma$ are parameters estimated by a nonlinear least-squares fit weighted by the reciprocal of the $XCO_2$ uncertainty (Zheng et al., 2020). $k \cdot l + b$ represents the background $XCO_2$, while the other part of Eq. (3) represents a single Gaussian shaped $XCO_2$ peak (Nassar et al., 2017; Reuter et al., 2019). After removing the background, the $CO_2$ line density is derived from calculating the area under the fitted curve (Figure S1b). The cross-sectional $CO_2$ flux is estimated by multiplying the $CO_2$ line density by the wind component perpendicular to the OCO-2 or OCO-3 orbit direction at the peak position of the plume in m



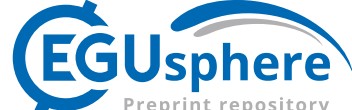

s$^{-1}$. Similarly, we allow the wind direction to rotate slightly to optimize the correlation between observations and the model simulation from Figure S1d to Figure S1c.

### 3.3. Detection of global power plant emission signals

In this study, we use the following steps to extract $XCO_2$ anomalies of global power plants for the estimation of their $CO_2$ emissions:

(1) Detect all satellite overpass data within a 0.25° radius around each power plant, and intercept all observations within the latitude range of ± 0.5° around the nearest observation from the power plant as potential cases;

(2) Extract the maximum value point of $XCO_2$ within the latitude range of ± 0.1° around the nearest observation point

from the power plant, and take this maximum point as the center to retain the data within the latitude range of ± 0.04°, as the Gaussian peak of the plume, and only retain the cases where the number of observations with sufficient quality in this range is more than 10 to minimize the effect of missing data in the plume.

(3) Calculate the angle between the vector from the power plant location to the maximum $XCO_2$ point and the wind direction vector, and only retain the cases where the angle is less than 60°, to ensure that the $XCO_2$ enhancement is located in

the downwind direction of the power plant.

(4) Retain the cases where the $XCO_2$ value of at least 5 observations in the Gaussian distribution are greater than the average value of the data extracted in step (1) plus 2 times the standard deviation of the data to ensure that the $XCO_2$ enhancement is significant.

(5) Extract the case where the net enhancement of $XCO_2$ of at least 5 observations is greater than 1.5 ppm. Here the

background is defined as 90$^{th}$ percentile of the data extracted in step (1).

(6) Finally, the automatically identified cases of power plant plumes were further screened visually. Four examples of cases which were rejected after visual inspection is shown in Figure S10. The identification of enhanced signals seen in the OCO data as resulting from a power plant outside the swath of OCO is further validated by using TROPOMI NO2 images for cases where a clear TROPOMI plume is available. The entire plumes observed by TROPOMI shown in Figure S8 and S9

shows that the association of the enhancement observed by OCO with the power plant was done correctly by the procedure. For the SAM data of OCO-3, only data in the same scan line are considered.

### 3.4. Uncertainty analysis and validation

(1) Emission uncertainties

The uncertainty estimates of this study are determined by three variables which are assumed to be uncorrelated. The total

uncertainty is calculated by error propagation as

$$\varepsilon_{Emission} = \sqrt{\varepsilon_{XCO2}^2 + \varepsilon_{wind}^2 + \varepsilon_{background}^2} \ , \qquad (4)$$



where each uncertainty is derived from the standard deviation of an ensemble approach. The uncertainty related to the wind $\varepsilon_{wind}$ is calculated from an ensemble of emission estimates based on WERA, WMERRA and WPBL. There are several possible approaches to determine the background. Hakkarainen *et al.* (2019) used the daily median of all $XCO_2$ within the

latitude range 10° band as the background to extract the $XCO_2$ anomalies. Nassar *et al.* (2017) determined background region from manual selection of observations outside the plume. Zheng *et al.* (2020) fitted the along-track observations by the sum of a Gaussian function and a linear function, where the linear part defined the background. We use a simple and automated way by calculating the percentile of the area defined in Sect. 3.3, step (1) and determining the background by taking the average value of all data below the percentile level. Here the background uncertainty $\varepsilon_{background}$ is computed from the spread in

emission estimates using the 75th, 80th, 85th, 90th percentile to define the background values. This range of percentiles leads to the smallest difference with the reported emissions, as shown in Figure S2. The $XCO_2$ uncertainty $\varepsilon_{XCO2}$ on the derived emission is computed by perturbing the original XCO2 data with the uncertainty of the retrieval as provided in the OCO-2 and OCO-3 data products.

(2)  Time-corrected hourly EPA reported values

The $CO_2$ emissions released by the power plant are transported to the satellite overpass location by the wind, and are detected as $XCO_2$ enhancement. EPA reports the emission value of the power plant on an hourly basis. When comparing emission estimates and hourly reported values, we need to consider the time lag between the moment when the emission is released at the stack and the moment it is detected downwind by the satellite. This time can be calculated from the distance between the power plant and the satellite crossing point and the wind speed. However, unlike the time-weighted reported

emission used in Nassar et al. (2021), we report the release time at the stack of power plant that the detected plume is only affected by the moment when it is generated. Therefore, we produce time-corrected hourly reported values at the time of the emissions seen by the satellite overpass instead of reporting the hourly emission values closest to the overpass time.

## 4.  Results

### 4.1.  Comparison of estimated emissions using hourly monitoring values

Figure 1a shows an example of a power plant emission plume in the satellite observations. The Jeffrey Energy Center power plant in Kansas was in operation at about 1:30 local time on October 30, 2020, and its $CO_2$ plume was captured by the downwind track of OCO-2. The local enhancement appears as a peak in latitude direction which cross-section is well approximated by a Gaussian (Fig. 1b). For the entire U.S., we analyzed the 1,284 plants reported by the EPA, of which 347 were excluded because of nearby city emissions. A total of 9,950 OCO-2 and 13,427 OCO-3 tracks were recorded within a

0.25° radius of these power plants. We used observations in the latitude range of ± 0.5° around the $XCO_2$ maximum for all tracks (the range shown in Figure 1a) and performed a visual selection to identify cases of enhancement from plumes of isolated



power plants like in Figure 1. The screening criterion is able to select a clear plume profile in the XCO₂ observations downwind of the power plant, such as in Figures 1e-1h, while other cases are rejected due to insignificant XCO2 enhancement, missing data and emission source cluster interference, such as in Figure S10. In the end, we arrived at 50 cases where the power plant

was operating and the emission plume crossed the satellite-track, including 30 cases from OCO-2 and 20 cases from OCO-3. When the distance between two adjacent power plants does not exceed the range of 1 pixel of the satellite, it was considered as a single isolated emission source, and their names are connected with commas (Table 1).

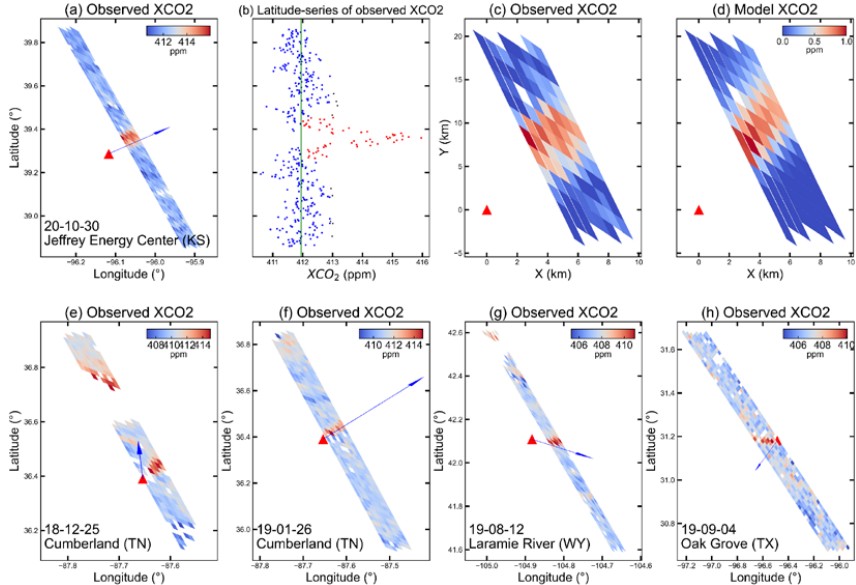

**Figure 1.** Estimation process of power plant emissions (a-d). (a) CO₂ plume of Jeffrey Energy Center power plant on October 30, 2020. (b) Change of XCO₂ in latitude direction from (a). The background value determined from the average of the observations below the 90th percentile (green line), background points (blue), and plume points (red). (c) Zoom-in of Figure (a) to the area of our simulation. (d) The simulated normalized XCO₂ enhancement for the same region by the GPM. (e-h) show cases of other power plant emission signals. The blue arrow represents the wind vector halfway the height of the PBL. The wind speeds in a, e, f, g and h are 2.9, 1.8, 6.1, 2.6 and 3.4 m s⁻¹,
respectively.

Previous studies used various choices of wind information to approximately account for the plume spreading, such as the wind speed at 250 m, the assumed average height of the chimney (Nassar et al., 2017, 2021; Chevallier et al., 2020), or at 31 m (Chevallier et al., 2022), or the average wind speed of the pressure layer near the ground as an approximation (Zheng et al., 2020; Hakkarainen et al., 2021), or calculate an effective wind (Varon et al., 2018; Reuter et al., 2019; Hakkarainen et al.,
2021). In this study, we compared the estimated CO₂ emission results driven by WERA5 (Figure S3), WMERRA (Figure S4) and WPBL used for the GPM method. The results show that the emission estimate obtained using WPBL give better results than the other two wind options. The correlation coefficient R of the estimated emission and time-corrected reported US EPA emission of the 50 cases of isolated power plants are 0.35, 0.28, and 0.14, for WPBL, WERA and WMERRA respectively (Figure 2a, Figure S3a, Figure S4a). These 50 cases contain multiple observations of 22 isolated power plants. For some power



plants, we have multiple observation days. For these cases we have averaged the results. As shown in Figure 2b, the correlation

coefficient R of the averaged estimated emission and the reported emission are 0.63, 0.44, and 0.22, respectively, corresponding

to WPBL, WERA and WMERRA (Figure 2b, Figure. S3b, Figure S4b). The improved correlation illustrates the large fraction

of randomness in the emission retrieval uncertainty (Chevallier et al. 2022). For the sum of the estimates of the repetitive cases

of each power plant, we obtained a better correlation of 0.93, 0.89, 0.73, respectively, corresponding to WPBL, WERA and

WMERRA respectively (Figure 2c, Figure S3c, Figure S4c). Therefore, we decided to use WPBL for the estimation of power

plant emissions.

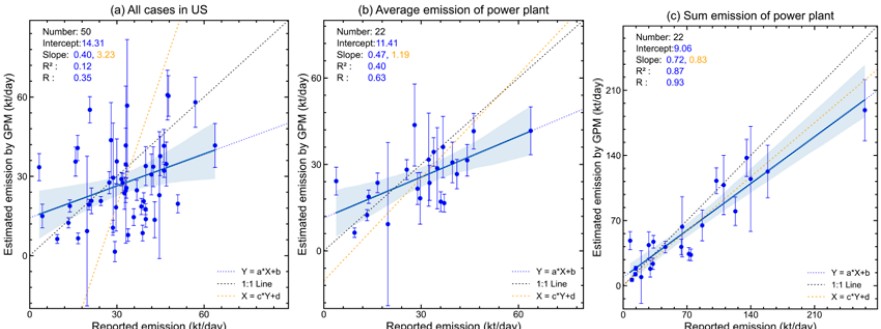

**Figure 2.** The emission estimation results by the GPM of all cases with WPBL are compared with the time-corrected hourly reported value from EPA (a), the average value of emission estimation results of each power plant is compared with the reported value (b), and the sum of emission estimation results of each power plant is compared with the reported value (c). The vertical error line is the total uncertainty of estimated emissions. The yellow and blue dashed lines are the fitted lines with the x-axis and y-axis swapped.

In Figure 3, we compare the GPM method and the cross-sectional flux method driven by WPBL. Among them were two

cases without a result from the cross-sectional flux method because of invalid fitting. It shows that the estimates from the

cross-sectional flux method fluctuate much more than the estimates from the GPM method. This is mainly because the orbits

of the OCO-2 and OCO3 satellites are not perpendicular to the plume, and the final step of the cross-sectional flux method

uses the wind field component normal to the orbit multiplied by the line density to estimate the flux, while the GPM method

derives the posteriori emission by a linear least squares fit between observed and simulated enhancements. Moreover, the

OCO-2 and OCO-3 observations do not sample the entire emission plume, as shown in Figure 1f, but just the part of the plume

cross section within the narrow width along the orbit. Hakkarainen *et al.* (2021) also found that the estimates from the cross-

sectional flux method fluctuated greatly. Therefore, we decided to use the GPM method for the estimation of global power

plant emissions. When compare plant level estimated emission with reported emission by the U.S. Energy Information

Administration (EIA) from fuel consumption records (https://www.eia.gov/electricity/data/emissions/), Figure S6 shows that

the correlation between estimated emission and reported annual emission from EIA (R = 0.39) is lower than that from EPA,

although the reported annual emissions from EPA and EIA reveal a good correlation (R = 0.80). Figure 4 shows that, due to

strongly hourly variations in the power plant emission, the satellite overpass time is not always representative for the annual





emission of a power plant. Note that EIA reports the yearly-mean emission based on the annual fuel consumption of the power

plant which will differ from the emission at the satellite overpass time.

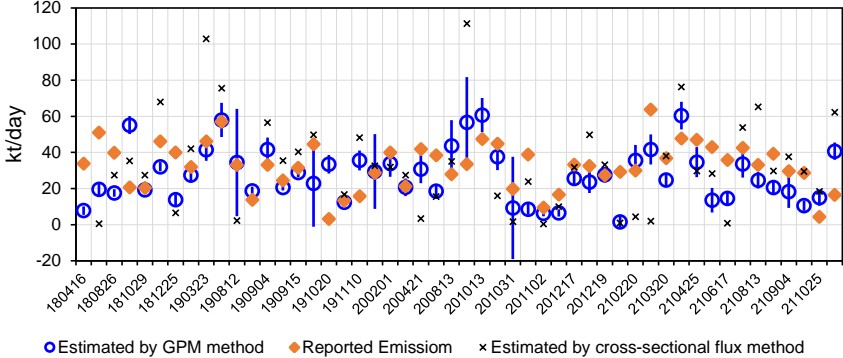

**Figure 3.** Emission values of U.S. power plants estimated with the GPM (blue circles) and cross-sectional flux (black crosses) methods
compared to the time-corrected reported values (orange diamonds).

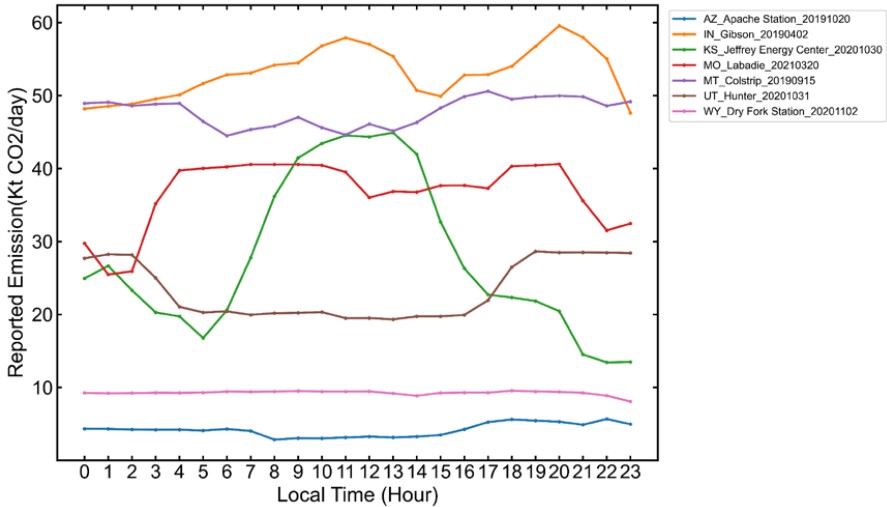

**Figure 4.** Hourly emission variation of 7 randomly selected power plants and dates from hourly US EPA data. The name of each curve
consists of the name of the state, the name of the power plant and the YYYYMMDD day.

   Table 1 lists the average estimated emissions for 22 power plants and the estimated uncertainty caused by the background,

$XCO_2$ and wind field. The deviation of the estimated emissions and reported emissions varies between 0.47 and 22.11 kt/day,

and the total uncertainty varies between 1.65 and 28.32 kt/day. Among the three uncertainty components, the uncertainty

caused by the wind field is the highest. From 2018 to 2021, for the $XCO_2$ archived data, there are 6 cases found for the Jeffrey

Energy Center power plant (KS) and 4 cases for the Prairie State Generating Station power plant (IL), Colstrip (MT),

Cumberland power plant (TN) and Oak Grove (TX). For a few power plants, we found that the time variability of estimated

and time-corrected hourly EPA reported emissions from multiple observation cases of power plant display a good consistency



(Figure S5), such as Gibson (IN) and Labadie power plants (MO). Excluding the two power plants whose uncertainties are greater than the estimated emissions, the uncertainties of the other cases are within 8 % to 51 % of the reported emissions.

**Table 1**. The average estimated emissions, reported emissions, uncertainty components, and number of observations from US power plants.

| Name | Reported emission (kt/d) | Estimated emission (kt/d) | Uncertainty of background (kt/d) | Uncertainty of XCO2 (kt/d) | Uncertainty of wind (kt/d) | Total uncertainty (kt/d) | Number of observations |
|---|---|---|---|---|---|---|---|
| James H Miller Jr (AL) | 63.8 | 41.7 | 1.7 | 1.0 | 8.1 | 8.3 | 1 |
| Apache Station (AZ) | 3.8 | 24.2 | 2.4 | 2.4 | 3.4 | 4.8 | 2 |
| Arlington, Mesquite, Redhawk Facility (AZ) | 13.3 | 12.4 | 0.8 | 0.3 | 1.7 | 1.9 | 1 |
| Prairie State Generating Station (IL) | 25.6 | 28.2 | 1.8 | 0.2 | 2.9 | 3.5 | 4 |
| Gibson (IN) | 36.7 | 36.0 | 1.6 | 1.2 | 10.3 | 10.6 | 3 |
| Jeffrey Energy Center (KS) | 44.2 | 31.4 | 1.5 | 0.4 | 5.1 | 5.5 | 6 |
| Iatan (MO) | 28.9 | 21.6 | 3.2 | 1.5 | 2.9 | 5.5 | 3 |
| Labadie (MO) | 41.0 | 26.7 | 1.9 | 0.8 | 4.5 | 5.1 | 3 |
| Colstrip (MT) | 35.0 | 28.7 | 1.2 | 0.5 | 13.9 | 14.1 | 4 |
| Gerald Gentleman Station (NE) | 29.7 | 18.3 | 1.3 | 0.4 | 8.9 | 9.0 | 1 |
| Four Corners Steam Elec Station (NM) | 16.6 | 23.6 | 2.3 | 0.5 | 2.5 | 3.4 | 2 |
| Cardinal (OH) | 37.1 | 16.6 | 0.9 | 0.6 | 2.9 | 3.1 | 2 |
| Conemaugh, Seward (PA) | 46.2 | 41.5 | 2.9 | 1.5 | 5.2 | 6.2 | 1 |
| Cumberland (TN) | 33.9 | 34.3 | 2.2 | 0.4 | 4.4 | 5.0 | 4 |
| Harrington, Nichols station (TX) | 28.0 | 43.7 | 2.2 | 0.6 | 14.0 | 14.2 | 1 |
| Oak Grove (TX) | 39.6 | 30.7 | 2.2 | 1.2 | 6.5 | 7.1 | 4 |
| Parish, Carbon-Capture, Brazos Energy (TX) | 36.1 | 17.1 | 1.1 | 0.8 | 3.1 | 3.4 | 2 |
| Sam Seymour (TX) | 32.6 | 23.6 | 1.3 | 0.0 | 5.9 | 6.1 | 1 |
| Hunter (UT) | 19.7 | 9.3 | 0.5 | 0.3 | 28.3 | 28.3 | 1 |
| Intermountain (UT) | 13.8 | 18.8 | 1.1 | 0.7 | 1.9 | 2.3 | 1 |
| Dry Fork Station (WY) | 9.5 | 6.3 | 0.6 | 0.1 | 1.5 | 1.6 | 1 |
| Laramie River (WY) | 32.3 | 31.7 | 1.5 | 0.2 | 15.9 | 16.1 | 2 |

### 4.2. Detection and estimation of global power plant emission signals

Figure 5 shows the number of cases retained for each processing step of automatic detection of global power plant emission signals using the GPPD. We obtained 1387 days of $XCO_2$ observation data from OCO-2 from January 2018 to December 2021 and 766 days of $XCO_2$ observation data from OCO-3 from August 2019 to November 2021. For 8660 power plants in the world, all tracks from OCO-2 and OCO-3 were scanned near the power plants. The number of cases with more than 10 observations (step 2 in Sect. 3.3) near the gaussian peak is 39365 and 42932 for OCO-2 and OCO-3. 40.71% of the $XCO_2$ cases are located in the downwind direction of the power plant. Among them, 24.94% of cases contain at least 5 observations that are significantly enhanced relative to the background. Among those 518 and 804 plume observations with at least 5 observation points from OCO-2 and OCO-3 have net enhancement exceeding 1.5 ppm. Finally, through visual selection




(step 6 in section 3.3), 83 and 23 cases from OCO-2 and OCO-3 were considered as plumes from isolated power plants,

respectively.

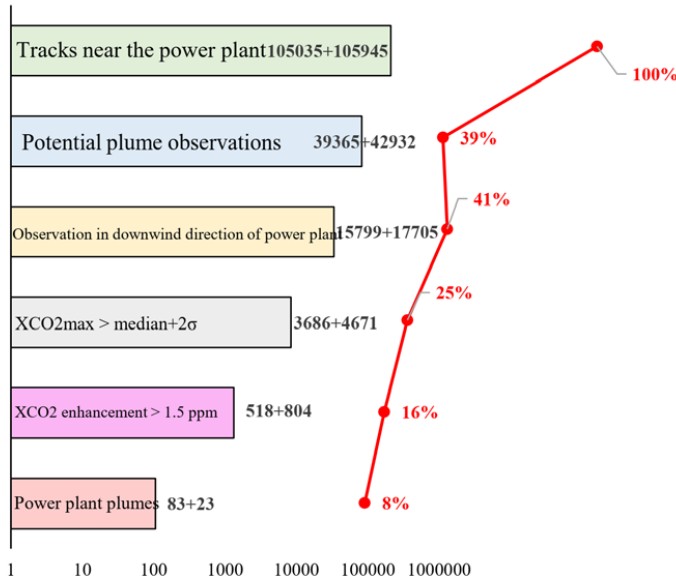

**Figure 5.** Statistics of the number of OCO-2/3 XCO2 observations respectively in each processing step. The red line represents the proportion of the selected observations in the previous step.

Figure 6 shows the estimated $CO_2$ emissions of 106 global power plant cases calculated by the GPM method using WPBL.

The estimated emissions of these power plants range from 3.2 to 109.0 kt/day., The percentiles of the estimated emissions of $25^{th}$, $50^{th}$ and $75^{th}$ are 19.9, 32.1 and 52.6 kt/day respectively, and the uncertainty range is 1.2~62.6 kt/day. Figure 6a shows the location of these power plants and their emissions indicated by circle size and color. Figure 6b shows the sum of estimated emissions for all found observations at each power plant. The gray vertical lines are an indication of the uncertainty of the estimated emissions. Furthermore, we calculated the correlation between the integral of the observed and simulated $XCO_2$

enhancement from equation (1) and (2) in the latitude direction. Figure S7 shows a correlation coefficient of 0.56 for observed and simulated enhancements of global power plant cases.



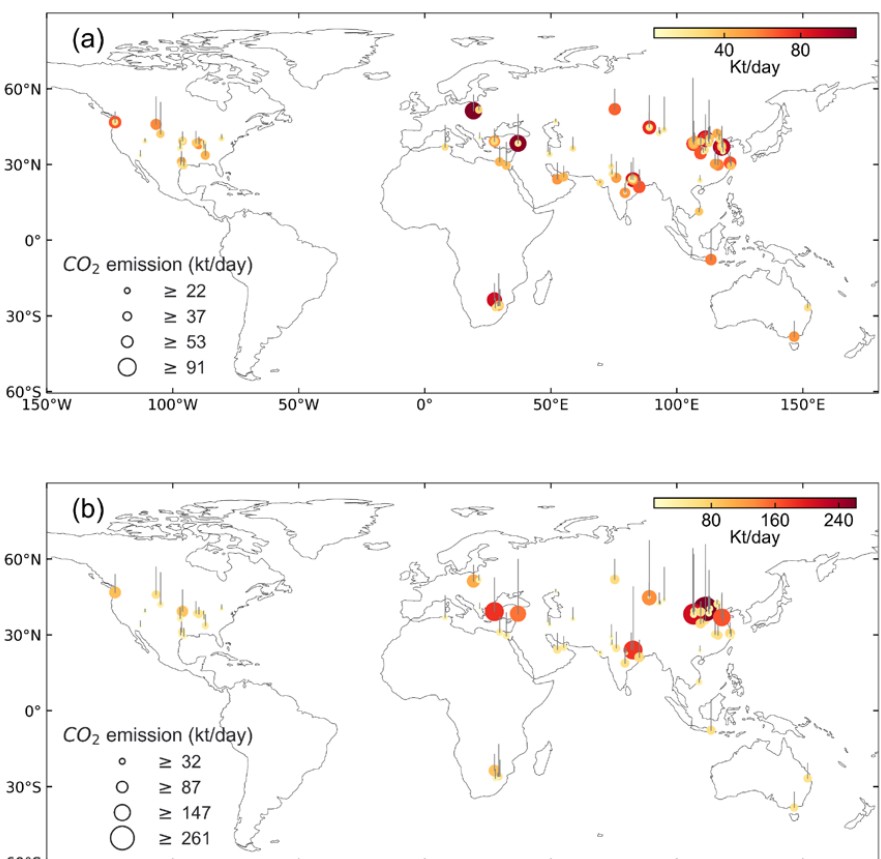

**Figure 6.** The detected global power plant emissions with emission estimation results (a), and the sum of emission estimation results of all
found observations at each power plant (b). The color and size of the circles indicate the estimated emission.

The detection algorithm reduces four-year global $XCO_2$ data to only 106 cases of 78 unique power plants. A large number
of power plant emission cases have been discarded due to insignificant $XCO_2$ enhancements of less than 1.5 ppm, not enough
valid observations in the plume and also by the visual check. We compare the estimated emissions with the carbon emission
inventory EDGAR V6.0 for the power sector in order to understand the magnitude of the emission of detected cases. The
estimated emissions of detected power plants are counted by year and country (Figure 7). When assuming constant emissions
of power plants, the sum of estimated emissions of all power plants would be extrapolated to $1522 \pm 501$ Mt/year, accounting
for about 17% of the all power sector emissions of countries showed in Figure 7b according to EDGAR in 2018. The estimated
emissions from the few observations in 2018, 2019, 2020 and 2021 account for only 2%, 5%, 6% and 4% respectively of all
power sector emissions of countries showed in Figure 7a in 2018. The top three countries in terms of detected estimated
emissions of power plants are China, United States and India. This illustrates the fact that OCO-2 and OCO-3 are only capable
to see a fraction of the emitted $CO_2$ emissions due to the limited spatial coverage of the instrument and the often cloudy
conditions during observation.





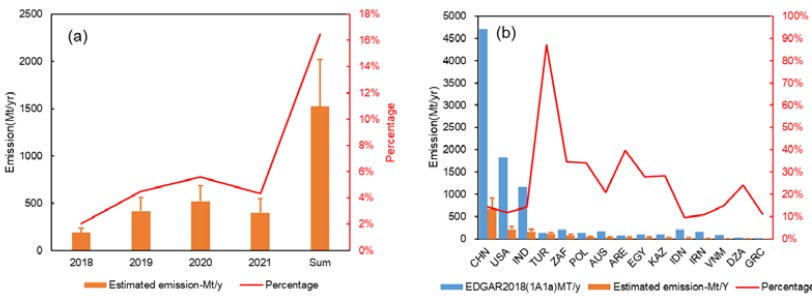

**Figure 7.** Estimated annual emissions of the detected global power plants, also shown as percentage of the total reported emissions of global power plants. (a) The red curve shows the proportion of annual estimated emissions to the total emissions of all countries with observations in 2018 (from EDGAR2018 V6.0 1A1a). (b) The red curve in the right figure shows the percentage of estimated emissions in comparison to the country total power plant emissions according to the inventory.

## 5. Summary and Conclusions

In this study, we compared two widely used methods for estimating point source emissions of power plants: the Gaussian plume model method and cross-sectional flux method. We applied the two methods to carefully selected power plant plumes in the United States observed by OCO-2 and OCO-3. The accuracy of the two methods is validated with time-corrected hourly reported emissions from EPA. We found that the cross-sectional flux method has a larger variability than the GPM method because it makes use of the wind field component normal to the orbit. We used the Gaussian plume model method to evaluate the impact of three kinds of wind field data sets (WPBL, WERA and WMERRA) on the accuracy of emission estimates of isolated power plants. The results show that, for a single case, the correlation between reported emission and estimated emission driven by WPBL is the highest. When there are multiple observations of the same power plant, the correlation between the average and total estimated emissions of the power plant and the reported emissions is significantly improved, from $R^2 =$ 0.40 to 0.87. No matter what kind of wind field data is used, the Gaussian plume model has a high correlation $R^2$ of the total emissions from power plants from multiple observations, which is above 0.5. In general, obtaining more observation data from more instruments can significantly reduce the uncertainty of estimated emissions of power plants.

Once having selected the best emission estimation method for isolated power plants, we applied this simple and fast method globally. We developed a procedure to automatically detect the emission signals of power plants, and, after a visual selection, obtained 106 global power plant emission observations of 78 power plants.

Unlike continuous imaging satellites, OCO-2 and OCO-3 scans cover a very limited part of the Earth's surface on a daily basis. By removing the cloud impact and only extracting the downwind emission plumes of power plants, the available observations are further reduced. The extremely limited number of cases from the existing satellites make it impossible to capture the time variability of power plants, whether diurnal or seasonal. In addition, only isolated emission hotspots are estimated here to avoid the impact of adjacent emission sources.

With the future increase of observation sensors with improved spatial-temporal resolution, such as the planned GeoCarb



of NASA and Carbon Dioxide Monitoring Mission (CO2M) of ESA, the probability of observing a $CO_2$ plume will greatly
increase. The abundant observation data obtained by the new generation of satellites will contribute to the monitoring of power
plant emissions worldwide. The emissions of power plants in a background of other emission signals may also be monitored
due to high-resolution observations and increased swath width, and the uncertainty of the estimated emissions of power plants
will further decrease.


*Author contributions.* Conceptualization and methodology: X.L., R.vd.A, J.D., F.C., Z.L., and P.C.; Data processing: X.L.,
H.E., Z.D., Y.G., X.S., X.N., D.H., and X.D.; Model simulation: X.L.; Formal analysis: X.L., R.vd.A, J.D.; Writing—original
draft: X.L.; Writing—review and editing: All authors; Visualization: X.L.; Supervision, project administration, funding
acquisition: Z.L.

*Competing interests.* The authors declare that they have no competing interests.

*Acknowledgements.* This research received funding from National Natural Science Foundation of China (71874097, 41921005,
71904007 and 71904104). The support provided by China Scholarship Council (CSC) during a visit by Xiaojuan Lin to Royal
Netherlands Meteorological Institute (KNMI) is acknowledged.

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
