# Peer review of "Monitoring and quantifying CO2 emissions of isolated power plants from space"

_EGUsphere, 2022_

## Author Comment (AC1)

**Title:** Monitoring and quantifying CO2 emissions of isolated power plants from space
**MS No.:** egusphere-2022-1490

Below we reply to the reviewer comments point by point. The reviewer comments are shown in *italic*, and corresponding modifications and citations of the manuscript are quoted.

Referee #1

*(1) Lin et al. "Monitoring and quantifying CO2 emissions of isolated power plants from space" builds off previous work on quantifying power plant emissions using OCO-2 and OCO-3 observations together with models. It is good to see this effort toward development of a more systematic and automated method that leverages what has been demonstrated by others in past case studies. Furthermore, the comparison between the Gaussian plume method (GPM) and Integrated Mass Enhancement (IME) method is a useful investigation that highlights the importance of the satellite coverage and resolution and the different nature of CO2 and CH4 plumes since the conclusion differs from that based on high spatial resolution CH4 observations in the literature. Overall, this is a useful study that helps to bring the field a step closer to the implementation of an operational system for CO2 anthropogenic emission monitoring as planned for CO2M. Following some minor revisions related to the specific points below, I would recommend its publication.*

**Response:** We thank Referee #1 for the encouraging comments. All comments and suggestions have been considered carefully and addressed below.

*Specific Points*

*(2) Line 43-44: These are not really the primary references regarding the difficulty to achieve accurate and detailed consumption data*

**Response:** We have changed it in the revised manuscript, as follows:

"especially for developing countries (Olivier et al., 2017; International Energy Agency, 2019; European Commission, 2019; Gilfillan and Marland, 2021)".

*(3) Line 63: Reuter et al. (2019) derived emission estimates for power plants, urban areas and wild fires*

**Response:** We have changed it in the revised manuscript, as follows:

"Reuter *et al.* (2019) used a few co-located regional enhancements of XCO2 and NO2 observed by OCO-2 and TROPOMI respectively to derive emission estimates for power plants, urban areas and wild fires".

*(4) Line 66: Nassar et al. (2022) https://www.frontiersin.org/articles/10.3389/frsen.2022.1028240/full is a key OCO-3 example worth mentioning*

**Response:** We add the reference of Nassar et al. (2022) in the revised manuscript, as follows:

"Nassar *et al.* (2017, 2021, 2022) extended the approach and applied it in backward mode in order to quantify $CO_2$ emissions from individual power plants using OCO-2 and OCO-3 $XCO_2$ data".

*(5) Line 71: Schwandner et al. 2017 is not the best choice of reference. Although the paper mentions power plants, it really focuses on XCO2 enhancements in an urban area (later understood to be topography related biases), while the only emission estimate is of volcanic emissions from one cloudy overpass*

**Response:** We have removed it in the revised manuscript.

*(6) Line 74: "manually-selected" is perhaps a better descriptor than "hand-picked" (slang)*

**Response:** We have corrected it as "manually-selected" in the revised manuscript,

*(7) Line 79: Intermittency of U.S. sources has previously been studied by Hill and Nassar (2019) https://doi.org/10.3390/rs11131608 and Velazco et al. (2011) www.atmos-meas-tech.net/4/2809/2011/, so these two past studies should be cited.*

**Response:** We have added the reference of Velazco et al. (2011) and Hill and Nassar (2019) in the revised manuscript, as follows:

"Velazco *et al.* (2011) quantified errors of power plant annual emission estimates by a hypothetical CarbonSat constellation. Hill and Nassar (2019) assessed pixel size and revisit rate requirements for monitoring power plant CO2 emissions from space".

*(8) Line 97: "≤ 1.29 x 2.25 km2" (It is worth noting that this is the maximum footprint size, since it is usually smaller due to solar angle and viewing geometry)*

**Response:** We have corrected it in the revised manuscript.

*(9) Line 97: "~52" degrees is recommended since the value can be exceeded by a few tenths of a degree in some cases*

**Response:** We have corrected it in the revised manuscript.

*(10) Line 111: daily global coverage before loss of data due to clouds*

**Response:** The tropospheric NO2 data of TROPOMI has daily global coverage, where each observation is having a quality factor. Depending on the application a (cloud) filtering can be applied to the original data. Therefore, we would like to keep this phrasing. In line 114 the filtering for clouds is mentioned.

*(11) Line 119: This EPA link has annual power plant emission data, but is it the correct link for the hourly data too?*

**Response:** We have updated the link in the revised manuscript, as follows: "https://www.epa.gov/airmarkets/power-sector-emissions-data".

*(12) Line 257: Nassar et al. 2021 used the assumed height of the chimney plus an assumed 250 m for typical plume rise above the stack height*

**Response:** Thanks. We have changed it in the revised manuscript, as follows:

"Previous studies used various choices of wind information to approximately account for the plume spreading, such as the wind speed at the assumed average height of the chimney (250m, Nassar et al., 2017; Chevallier et al., 2022), or the assumed height of the chimney plus an assumed 250 m for typical plume rise above the stack height (Nassar et al., 2021), or at …".

*(13) Line 295: For clarify, it would be helpful to specify that the x-axis is labelled with YYMMDD.*

**Response:** Thanks. We have updated Figure 3 in the revised manuscript.

*(14) Line 374: Should revise language about GeoCarb as it has recently been cancelled by NASA.*

*Line 375: CO2M is a Copernicus mission with ESA and EUMETSAT involvement*

**Response:** We have corrected it in the revised manuscript based on the comments of Reviewer 1 and Reviewer 2, as follows:

"such as the planned European Carbon Dioxide Monitoring Mission (CO2M) and the Japanese Global Observing Satellite for Greenhouse gases and Water cycle (GOSAT-GW)".

---

## Author Comment (AC2)

**Title:** Monitoring and quantifying CO2 emissions of isolated power plants from space
**MS No.:** egusphere-2022-1490

Below we reply to the reviewer comments point by point. The reviewer comments are shown in *italic*, and corresponding modifications and citations of the manuscript are quoted.

Referee #2

*(1) The study estimates CO2 emission of power plant using OCO-2 and OCO-3 observations using the Gaussian plume inversion and cross-sectional flux method with different input parameters. The methods are tested for U.S. power plant for which bottom-up reports of hourly emissions are available and afterwards applied globally. The paper well written but some aspects on the method are unclear. I would recommend publication following a revision based on the general and specific comments below:*

**Response:** We thank Referee #2 for the encouraging comments. All comments and suggestions have been considered carefully and addressed below.

*(2) Background: For Gaussian plume model method please describe already in L153ff how you calculate the background. In L200, you write that the 90th percentile was used, which seems to be based on tests with different percentiles with the aim to minimize the difference between estimated and reported emissions (L220, Figure S2). The choice of the percentile (60-99th) will mostly result in a bias in the estimated emissions, which might be caused by the background, but can also be the result of other systematic errors in other parameters (e.g. effective wind speed). Therefore, how does this choice of the background agree with the background that you compute with the cross sectional flux method (L175)? Would a different background affect your conclusions on the best approach for computing the effective wind speed?*

**Response:** The different backgrounds do not affect the conclusions of this study regarding the comparison of three wind fields, because the difference in background obtained by these two methods is very small (Fig. S11a), with a maximum of 0.86 ppm and a minimum of 0.004 ppm (Fig. S11b). Under the two background calculation methods, the GPM method has good consistency in the estimation results driven by three wind field (Fig. S11c-e). With the background computing by Eq. (3), the conclusion that estimated emissions have better accuracy using the WPBL is still valid (Fig. S12).

For all 50 cases, the difference between the background calculated by the 99 percentile and the background calculated by the 60 percentile ranges from 0.23 to 0.77 ppm. The standard deviation of the background calculated for the 9 percentile bins for each case ranges from 0.08 to 0.26 ppm.

In this study, the background for the Cross-sectional flux method (CFM) is determined by fitting of Eq. (3), while the background for the Gaussian plume model method (GPM) is determined by the $90^{th}$ percentile. However, more importantly, the Gaussian fitting in Eq. (3) may fail for some cases (e.g., Fig.S8a, S9a), but these cases can be estimated by the GPM method. The GPM method can simulate the enhancements of XCO2 at any location downwind of the emission source in the two-dimensional plane (Fig.1d), which cannot be well described by the Gaussian fitting. Therefore, in this study, the two methods use two different background calculation methods.

In response to the sensitivity regarding the background information mentioned above, we have made changes to the manuscript in L280-286: "In this study, the background for the cross-sectional flux method is determined by fitting of Eq. (3), while the background for the Gaussian plume model method (GPM) is determined by the 90th percentile. The difference in background obtained by these two methods is very small (Fig. S11a), with a maximum difference of 0.86 ppm and a minimum of 0.004 ppm (Fig. S11b). Under the two background calculation methods, the GPM method has good consistency in the estimation results driven by three wind field (Fig. S11c-e). With the background computing by Eq. (3), the conclusion that estimated emissions have better accuracy using the WPBL is still valid (Fig. S12)." and the supplement in L25-27 and L74-85. We have also added the following Figure S11, S12 to the supplement:

[Figure]

**Figure S11.** Comparison of two methods of computing the background. The background constant b from Eq. (3) is in good agreement with the background calculated by using the 90th percentile (a), and their difference is small (b). Under the two background calculation methods, the GPM method has good consistency in the estimation results driven by WPBL (c), WERA (d), and WMERRS (e) wind fields, respectively.

[Figure]

[Figure]

**Figure S12.** The conclusion that estimated emissions have better accuracy under the WPBL is still valid when using the background calculated by the method of Eq. (3). These three panels are based on WPBL (a1-a3), WERA(b1-b3), and WMERRA(c1-c3).

*(3)   Wind: The evaluation of the different wind products in your study is inconsistent. You use winds from ERA-5 (0.25°), MERRA-2 (0.5°x0.625°) and high-resolution ECWMF forecast (resolution not mentioned). You use the wind speed at the center of the PBL for the ECMWF forecast. However, for ERA-5 and MERRA-2, you take 10-m winds multiplied by the empirical scaling factor of 1.4 from Varon et al. (2018). When you compare the impact of the different wind products on the estimated emissions, it is not possible to identify if the different performances are caused by differences between the products or the different computation of the final product (scaling factor or wind at half PBL height). To analyse this better, I suggest comparing all datasets using both the 1.4-factor and the wind at half the height of the PBL. Note that the scaling factor of 1.4 is derived for CH4 plumes measured by high-resolution satellites, which are inherently different to CO2 plumes from power plants measured by OCO-2. Therefore, using the value might be not the best approach, even it is true that it has been used in previous studies "for convenience" (Reuter et al. 2019).*

**Response:** The resolution of ECMWF (WPBL) is $0.25° \times 0.25°$ (added in L140).

In this study, we compared the effective wind computed from ERA5 and MERRA2 10 m wind speed with the wind at half the height of the PBL. To some extent, the wind at half the height of the PBL is also a representation of the average state of the plume and an effective wind. Finally, we found that the results using the wind at half the height of the PBL were the best (Fig. 2), which suggests that it represents the spreading of $CO_2$ plumes in the vertical direction more accurately (Fig. 2, Fig.

S3, Fig. S4). The reason why the results using MERRA2 were the worst (Fig. S4) is due to its low resolution, which cannot provide precise wind information for emission sources. Here, ERA-5 and operational forecast of ECMWF are very similar and assumed to perform in the same way. We didn't use the MERRA2 wind at half the height of PBL because of its lower resolution.

Varon *et al.* (2018) calculated an effective wind speed $U_{eff}$ (Eq. S1-2) representing the average state of the vertical variability of the wind, based on the known Q and C, and found a multiple relationship (1.3 ~ 1.5) between $U_{eff}$ and the 10 m wind speed. Hence, subsequent studies used a scaling factor of 1.4 applied to the 10 m wind speed to represent the average state of the plume in the vertical direction (Reuter et al., 2019; Hakkarainen et al., 2021). Although this factor was derived for methane (lighter than CO2) plumes, it is a commonly used method.

$$Q = \int_{-\infty}^{+\infty} U(x, y)\Delta\Omega(x, y)dy, \qquad \text{(S1)}$$

$$Q = CU_{eff}, \quad \text{with} \quad C = \int_{-\infty}^{+\infty} \Delta\Omega(x, y)dy. \qquad \text{(S2)}$$

We changed the manuscript in L267~272:

"The correlation coefficient R of the estimated emission and time-corrected reported US EPA emission of the 50 cases of isolated power plants are 0.35, 0.28, and 0.14, for WPBL, WERA and WMERRA respectively (Figure 2a, Figure S3a, Figure S4a). The results show that the emission estimate obtained using WPBL give better results than the other two wind options, which suggests that it represents the spreading of CO2 plumes in the vertical direction more accurately (Fig. 2, Fig. S3, Fig. S4). The reason why the results using MERRA2 were worse (Fig. S4) is due to its low resolution, which cannot provide precise wind information for emission sources."

(4) *Uncertainties: You seem to compute the uncertainties using an ensemble approach with a rather small number of members (3 for wind and 4 for background) for computing reasonable statistics (see also previous comment on the wind). How large are the uncertainties of wind speed in m/s and background in ppm? How do these uncertainty estimates compare to estimated uncertainties in previous studies? How large is the uncertainty of the fitting parameters for the background in Equation 3? Finally, how large would be the uncertainties from the assumption and simplification of your methods such as the assumption of steady state conditions?*

**Response:**

*How do these uncertainty estimates compare to estimated uncertainties in previous studies?*

We added the uncertainty comparison in the revised manuscript L316-317: "The total uncertainty is comparable to the uncertainty of power plant emissions in previous studies, which ranged from 3.42 to 19.2 (Nassar et al., 2017; 2022)"

*How large is the uncertainty of the fitting parameters for the background in Equation 3?*

For all cases, the uncertainty of the background was only considered for the four percentile values ($75^{th}$, $80^{th}$, $85^{th}$, $90^{th}$ percentile). As mentioned in our response to question (2), the difference in background values calculated by the two methods is small (Fig. S11). The manually selected cases ensured the reliability of the isolated power plant observation signal, i.e., the background uncertainty is low.

*how large would be the uncertainties from the assumption and simplification of your methods such as*

*the assumption of steady state conditions?*

We think this is out of scope of our current paper and to be analysed in future research. We mentioned this in the revised manuscript of discussion L399-398, as follows:

"This study has only considered three sources of uncertainty. Future research may investigate additional sources, such as the assumption of steady state conditions and the plume rise, to better understand their impact on the results."

*How large are the uncertainties of wind speed in m/s and background in ppm?*

We added the uncertainty in the revised manuscript L317-318: "The uncertainty of wind speed is between 0.08 and 1.4 m s$^{-1}$, and the uncertainty of background varies between 0.03 and 0.1 ppm (Table S1)."

**Table S1.** The uncertainties of wind speed in m/s and background in ppm for each plant

| Name | Uncertainty of background (ppm) | Uncertainty of wind (m/s) |
|---|---|---|
| James H Miller Jr (AL) | 0.098 | 0.087 |
| Apache Station (AZ) | 0.039 | 0.235 |
| Arlington, Mesquite, Redhawk Facility (AZ) | 0.043 | 0.246 |
| Prairie State Generating Station (IL) | 0.063 | 0.238 |
| Gibson (IN) | 0.067 | 0.307 |
| Jeffrey Energy Center (KS) | 0.062 | 0.220 |
| Iatan (MO) | 0.076 | 0.355 |
| Labadie (MO) | 0.069 | 0.315 |
| Colstrip (MT) | 0.067 | 0.421 |
| Gerald Gentleman Station (NE) | 0.110 | 0.548 |
| Four Corners Steam Elec Station (NM) | 0.052 | 0.727 |
| Cardinal (OH) | 0.066 | 0.230 |
| Conemaugh, Seward (PA) | 0.054 | 1.491 |
| Cumberland (TN) | 0.066 | 0.940 |
| Harrington, Nichols station (TX) | 0.078 | 0.172 |
| Oak Grove (TX) | 0.077 | 0.366 |
| Parish, Carbon-Capture, Brazos Energy (TX) | 0.069 | 0.528 |
| Sam Seymour (TX) | 0.070 | 0.583 |
| Hunter (UT) | 0.070 | 0.219 |
| Intermountain (UT) | 0.051 | 0.209 |
| Dry Fork Station (WY) | 0.042 | 0.960 |
| Laramie River (WY) | 0.062 | 1.052 |

*Specific comments*

(5) *L171ff: You write here that you fit equation 3 to obtain parameters, k, b, A and sigma. Then, you write that you subtract the background to compute the line density. However, your parameter A is already the line density, so I don't understand why you need to calculate it again.*

**Response:** We use the Eq. 3 to fit the discrete observations along-track distance into a continuous curve, and further obtain the integral of the cross section (C in Eq. S2). The parameter A is one of the parameters to be fitted and not the line density. The line density (ppm·m) is the shaded area of Fig. R4 (the cross-plume integral), and then converted to g m$^{-1}$ through Eq. 2. Note that the definition of the parameter A

here is different from Kuhlmann et al. (2021)

$$V(x,y) = \frac{F}{\sqrt{2\pi} \cdot \beta \cdot (\frac{x}{1000})^{0.894} \cdot u} e^{-\frac{1}{2}(\frac{y}{\beta \cdot (x/1000)^{0.894}})^2} \quad , \qquad (1)$$

$$XCO_2 = V \cdot \frac{m_{air}}{m_{CO_2}} \cdot \frac{g}{P_{surf} - \omega \cdot g} \cdot 1000 \quad , \qquad (2)$$

$$f(l) = k \cdot l + b + \frac{A}{\sigma\sqrt{2\pi}} e^{[-(l)^2/2\sigma^2]} \quad , \qquad (3)$$

[Figure]

**Figure R4.** The blue shaded area is the line density. The peak is A/ (sigma*sqrt (2*pi)).

*(6) L182: It is not clear to me how you compute the wind here. Do you rotate the wind vector so that it points from the source location to the maximum in the OCO swath?*

**Response:** For the CFM, we use the same fitting of the wind direction as the GPM described in the previous paragraph: "The wind direction is allowed to rotate within a range of $\pm\,60°$ to account for errors in the wind data. The optimal wind direction is derived by maximizing the correlation coefficient between the simulated and the observed XCO2 enhancement". The end result is that the wind direction will roughly point towards the plume points (red points in Fig. 1b), possibly deviating from the maximum, because it is determined by all the red points together.

*(7) L235: You write that the peak is well described by a Gaussian [curve] in Figure 1b. However, no curve is shown in the figure.*

**Response:** Thanks. We have added the Gaussian curve in Figure 1b of the revised manuscript, as follows:

[Figure]

**Figure 1.** Estimation process of power plant emissions (a-d).

*(8) L240ff: You partly repeat the description of your method here, which seems unnecessary.*

**Response:** We agree that this part repeats the description of method. However, to better help readers understand the cases in Figure 1 and Figure S10, we explained how to distinguish and select clear plume cases and reject noise cases, so we prefer to keep it.

*(9) L261: Please discuss why WPBL provides better results than the other two options.*

**Response:** We have added why WPBL provides better results in the revised manuscript L267-272: "The correlation coefficient R of the estimated emission and time-corrected reported US EPA emission of the 50 cases of isolated power plants are 0.35, 0.28, and 0.14, for WPBL, WERA and WMERRA respectively (Figure 2a, Figure S3a, Figure S4a). The results show that the emission estimate obtained using WPBL gives better results than the other two wind options, which suggests that it represents the spreading of $CO_2$ plumes in the vertical direction more accurately (Fig. 2, Fig. S3, Fig. S4). The reason why the results using MERRA2 were worse (Fig. S4) is due to its low resolution, which cannot provide precise wind information for emission sources."

*(10) L265: Do you use the arithmetic average or the weighted average considering the uncertainty of the estimates?*

**Response:** We used the arithmetic average. We also tested the weighted average considering the uncertainty of the estimates and get the same conclusion that WPBL has the best performance shown in the following figures. We added an additional statement in the manuscript L275-277: "We also tested the weighted average considering the uncertainty of the estimates and reached the same conclusion that WPBL shows the best performance."

The weighted average considering the uncertainty of the estimates:

[Figure]

The arithmetic average (used):

[Figure]

*(11) L268/Figure 2: It surprises me that r² is so much higher for summing compared to averaging? Can*

*you explain why this is the case?*

**Response:**

When summing emissions for each plant, the data range is much higher than averaging. When a power plant has multiple observations, the difference between the sum of estimated and reported emissions for these observations is very small. Using the sum of the emissions is a non-standard approach. Generally one wants to compare similar characteristics of a process, in this case the typical (average) emissions per power plant.

However, with significant variations in number of collocations per powerplant, errors in average emissions vary considerably between power plants. Furthermore, emissions themselves vary considerably between power plants. As a result, more accurate and less accurate emission estimates are mixed in figure 2, which complicates reading and interpreting the plot.

Using the sum of emissions helps to visually discriminate between locations with more or fewer collocations. For the emission estimates themselves this does not matter: if the method is accurate the results should always align regardless of whether comparing averages or sums.

By displaying results from both approaches in one plot the reader can always compare the results from both methods.

*(12) L279: This relates back to my previous comment how you do calculate the normal wind for both methods. Are the difference between estimates and wind speed used in both method correlated? Another reason for deviations can be the method for computing the background.*

**Response:** The wind data finally used is the same for the GPM and CFM and is WPBL. The background fitted using Eq. (3) and the background calculated from percentiles differ very little (Fig. S11). More details about the error of the CFM method are shown in our response to question (16).

*(13) Figure 5: The red line is somewhat misleading, because without reading the caption one would assume that you could estimate emissions for 8% of all tracks near power plant, while in truths it is only 0.05%. I would strongly suggest removing it to avoid confusion.*

**Response:** We agree that the red line is confusing. We have removed the red line of Figure 5 in the revised manuscript.

*(14) L340ff: Does this number of 1522 Mt/a correctly accounts for observing the same power plant in different years or does this never happens? I am asking because the percentage numbers for the individual years add up exactly to 17, which would not happen if you estimate for the same power plant more than once.*

**Response:** For the same power plant, when there are multiple observation cases spanning multiple years, we directly take the estimated average of these cases. The 106 cases involved 78 unique power plants. Then we extrapolated the average emission (kt/day) of 78 power plants to the annual emission (Mt/year), and finally it was concluded that it accounted for 17% of the power sector emissions in 2018.

*(15) Figure 7b: It is difficult to see the bars for most countries. Maybe the figure can use a logarithmic scale on the y-axis.*

**Response:** We thank you for the suggestions. We have changed the Figure 7b with a logarithmic scale on the y-axis in the revised manuscript, as follows:

[Figure]

**Figure 7.** (b) The red curve in the right figure shows the percentage of estimated emissions in comparison to the country total power plant emissions according to the inventory.

*(16) L367f: The conclusion on the difference between cross-sectional flux and Gaussian plume method needs more explanations (see previous comment).*

**Response:** For the CFM method, by mass balance, the integral of the cross-section (Eq. S1) perpendicular to the downwind direction of the emission source multiplied by the wind speed of the emission source is the emission rate of the source. However, for the OCO-2 satellite with a narrow-width scanning mode, its orbit has a certain angle with the cross-section perpendicular to the plume, that is, the satellite orbit is not perpendicular to the wind direction of the emission source. Note although OCO-3 obtained plume data over a larger area in urban regions through multiple scans, we only selected one scan orbit as our analysis target. Therefore, an alternative method is to estimate that the emission rate is equal to the cross-sectional flux area multiplied by the component of wind perpendicular to the orbit. The maximum error in the instability of this method comes from the component of wind perpendicular to the orbit. If the satellite's trajectory and wind direction are completely perpendicular, the CFM method is feasible. This is the fundamental reason for the large error in estimating point source $CO_2$ emissions using the CFM method applied to OCO-2 satellite observations. On the other hand, GPM directly simulates XCO2 enhancement at any downwind position using the wind direction of the emission source, avoiding this issue and obtaining more stable results.

We added explanations in the revised manuscript L378-382: "We found that the cross-sectional flux method has a larger variability than the GPM method. This is because the emission rate from the cross-sectional flux method is equal to the cross-sectional flux area multiplied by the component of wind perpendicular to the orbit. The maximum error in the instability of this method comes from the component of wind perpendicular to the orbit. However, the GPM method directly simulates XCO2 enhancements at any downwind position using the wind direction of the emission source, avoiding this issue and obtaining more stable results."

[Figure]

*(17) L374f: Unfortunately, GeoCarb was recently canceled. CO2M is developed by ESA and EU with involvement from EUMETSAT and ECMWF. It is probably easiest to call it a "European mission". The Japanese GOSAT-GW should be mention here, as well.*

**Response:** Thanks. We have corrected it in the revised manuscript, as follows:

"such as the planned European Carbon Dioxide Monitoring Mission (CO2M) and the Japanese Global Observing Satellite for Greenhouse gases and Water cycle (GOSAT-GW)"."

*(18) Supplement: The resolution of the figures in the supplement is very low making it difficult to read the labels. In some cases, labels and units are missing (e.g., S9).*

**Response:** We thank you for the suggestions. We have changed the figures with higher resolution in the revised manuscript, as follows:

[Figure]

**Figure S8.** XCO2 and same-day NO2 concentration (0.025° × 0.025°) for three OCO2 cases.

[Figure]

[Figure]

**Figure S9.** XCO2 and same-day NO2 concentration (0.025° × 0.025°) for three OCO3 cases.

Hakkarainen, J., Szeląg, M. E., Ialongo, I., Retscher, C., Oda, T. and Crisp, D.: Analyzing nitrogen oxides to carbon dioxide emission ratios from space: A case study of Matimba Power Station in South Africa, Atmos. Environ.: X, 10, 100110, https://doi.org/10.1016/j.aeaoa.2021.100110, 2021.

Kuhlmann, G., Henne, S., Meijer, Y. and Brunner, D.: Quantifying CO2 Emissions of Power Plants With CO2 and NO2 Imaging Satellites, Frontiers in Remote Sensing, 2, https://doi.org/10.3389/frsen.2021.689838, 2021.

Nassar, R., Hill, T. G., McLinden, C. A., Wunch, D., Jones, D. B. A. and Crisp, D.: Quantifying CO2 Emissions From Individual Power Plants From Space, Geophys. Res. Lett., 44, 10045-10053, https://doi.org/10.1002/2017gl074702, 2017.

Nassar, R., Moeini, O., Mastrogiacomo, J.-P., O'Dell, C. W., Nelson, R. R., Kiel, M., Chatterjee, A., Eldering, A. and Crisp, D.: Tracking CO2 emission reductions from space: A case study at Europe's largest fossil fuel power plant, Frontiers in Remote Sensing, 3, https://doi.org/10.3389/frsen.2022.1028240, 2022.

Reuter, M., Buchwitz, M., Schneising, O., Krautwurst, S., O'Dell, C., Richter, A., Bovensmann, H. and Burrows, J. P.: Towards monitoring localized CO2 emissions from space: co-located regional CO2 and NO2 enhancements observed by the OCO-2 and S5P satellites, Atmos. Chem. Phys., 19, 9371-9383, https://doi.org/10.5194/acp-19-9371-2019, 2019.

Varon, D. J., Jacob, D. J., McKeever, J., Jervis, D., Durak, B. O. A., Xia, Y. and Huang, Y.: Quantifying methane point sources from fine-scale satellite observations of atmospheric methane plumes, Atmos. Meas. Tech., 11, 5673-5686, https://doi.org/10.5194/amt-11-5673-2018, 2018.

---

## Author Response (AR2)

**Title:** Monitoring and quantifying CO2 emissions of isolated power plants from space
**MS No.:** egusphere-2022-1490

Below we reply to the reviewer comments point by point. The reviewer comments are shown in *italic*, and corresponding modifications and citations of the manuscript are quoted.

Report #1

1.  *>> (2) For all 50 cases, the difference between the background calculated by the 99 percentile and the background calculated by the 60 percentile ranges from 0.23 to 0.77 ppm. The standard deviation of the background calculated for the 9 percentile bins for each case ranges from 0.08 to 0.26 ppm. [...]*

    *In response to the sensitivity regarding the background information mentioned above, we have made changes to the manuscript in L280-286: "In this study, the background for the cross-sectional flux method is determined by fitting of Eq. (3), while the background for the Gaussian plume model method (GPM) is determined by the 90th percentile. The difference in background obtained by these two methods is very small (Fig. S11a), with a maximum difference of 0.86 ppm and a minimum of 0.004 ppm (Fig. S11b). Under the two background calculation methods, the GPM method has good consistency in the estimation results driven by three wind field (Fig. S11c-e). With the background computing by Eq. (3), the conclusion that estimated emissions have better accuracy using the WPBL is still valid (Fig. S12)." <<*

    *Comment: Thank you for addressing this comment and for the additional analysis. A bias in the XCO2 background ranging from 0.23 to 0.77 ppm using different percentiles is quite large given that a 0.5 ppm bias in the background translates to about 10% bias in the estimated emissions for a XCO2 enhancement of 5 ppm. I would therefore not use the term "very small" to describe the result.*

    *Furthermore, you picked the 90th percentile for your study. Can you provide an argument for this specific value?*

**Response:** "very small" is referring to the difference between the two methods, which is on average rather small, but indeed the maximum in the difference is rather high. Thus, we adapted the text in L280-286: "The difference in background obtained by these two methods is on average small (Fig. S11a), but with a maximum difference of 0.86 ppm and a minimum of 0.004 ppm (Fig. S11b)."

In Fig. S2, we tested performance at 9 percentiles (60th, 65th, 70th, 75th, 80th, 85th, 90th, 95th, 99th). For each percentile, we calculated the coefficient of determination R2, root mean square error RMSE, and mean absolute error MAE between estimated and reported emissions. We found that for all cases, the error was smallest at the 90th percentile, so in the end we chose the 90th percentile. For the average emission and emission sum of each power plant, the error is smallest at the 75th percentile, so when we choose to calculate the background uncertainty, we choose 4 percentiles (75th, 80th, 85th, 90th). The relevant description is in L225-L228: "Here the background uncertainty $\varepsilon_{background}$ is computed from the spread in emission estimates using the 75th, 80th, 85th, 90th percentile to define the background values. This range of percentiles lead to the smallest difference with the reported emissions, as shown in Figure S2." We added the text in L280-281: "while the background for the Gaussian plume model method (GPM) is determined by the 90th percentile showing the lowest error for all cases in Figure S2a".

[Figure]

**Figure S2.** Different background results in different errors of estimated emissions and hourly reported emissions for isolated U.S. power plants. The errors of estimated emissions and hourly reported emissions by the GPM of all cases (a), the errors of the average value of emission estimation results of each power plant (b), and the errors of the sum of emission estimation results of each power plant (c).

2. >> (3) In this study, we compared the effective wind computed from ERA5 and MERRA2 10 m wind speed with the wind at half the height of the PBL. [...] Here, ERA-5 and operational forecast of ECMWF are very similar and assumed to perform in the same way. We didn't use the MERRA2 wind at half the height of PBL because of its lower resolution. <<

   Comment: Thank you for addressing my comment. I agree that using half the height of the PBL should give good estimates of the effective wind speed, because it provides you a reasonable estimate of the mean wind speed within the PBL, where the CO2 plume should be well mixed during OCO-2 overpass in the early afternoon (see also Brunner et al. 2023, https://doi.org/10.5194/acp-23-2699-2023).

   However, I think you miss my main issue with your comparison. You compare three model products in your study: (1) ERA-5, (2) MERRA-2 and (3) ECMWF forecast. In addition, you compare two methods for computing the effective wind: (A) 1.4x 10-m winds and (B) winds at half the PBL height. As a reader, I like to know which model and method is most suitable for estimating power plant emissions. However, it is not possible to get this information from your analysis, because you only compare three options: 1A, 2A and 3B. I don't think it is enough to assume that ERA-5 and ECMWF forecast are similar. I would actually assume that ERA-5 performs better than ECMWF, because a reanalysis should be better than an operational forecast. I therefore think it is necessary to compare the three products using the winds at half the PBL height to obtain an objective result.

   I am not convinced that the 1.4-factor is a general value that can be applied for computing the effective wind speed for OCO-2 CO2 observations of power plants. In fact, Reuter et al. (2019) only used this factor "for convenience", while Hakkarainen et al. (2021) derives a scaling factor based on the surface pressure at the Matimba power plant, which (coincidently) was consistent with Varon et al. (2018). Varon et al. (2018) derive their factor specifically for CH4 plume observations with GHGsat instrument (50 m resolution, 1-5% instrument precision). The factor is directly linked to the detection limit and pixel size of GHGsat, because they only integrate over the detectable width of the plume. It is therefore not possible to generalize their results to OCO-2 CO2 observations with lower spatial resolution (2 km) and different detection limit. Anyway, I don't think it is necessary to have a detailed discussions on this topic, because you already conclude that using the 1.4-factor results in worse performance than using the

*half-PBL value.*
*(1) ERA-5, (2) MERRA-2 and (3) ECMWF forecast*
*(A) 1.4x 10-m winds and (B) winds at half the PBL height*

**Response:** The comparison of option 1A and 2A is used to decide which meteorological field is performing better, ERA-5 or MERRA. This analysis clearly showed that MERRA is performing less, so that excluded the need for option 2A and 2B from the comparison. Subsequently, we conclude that the option (B) is better than option (A) and therefore 3A was no longer need checked.

ERA-5 is supposed to be better (for the past) than the operational ECMWF, but the differences in our case are small since we do not use any forecast of a few days ahead. Therefore, 1B and 3B are very similar and not compared, assuming that 1B is slightly better.

It would have been more thorough work if we had tested both 1B and 3B, but this would require a substantial amount of additional work and we prefer to leave this open for future research.

3. *>> (4) L316-317: "The total uncertainty is comparable to the uncertainty of power plant emissions in previous studies, which ranged from 3.42 to 19.2 (Nassar et al., 2017; 2022)" <<*
   *Comment: Please add units.*

**Response:** Thanks. We have added units in the revised manuscript, as follows:
"The total uncertainty is comparable to the uncertainty of power plant emissions in previous studies, which ranged from 3.42 to 19.2 kt/day (Nassar et al., 2017; 2022)."

4. *>> (4) "The uncertainty of wind speed is between 0.08 and 1.4 m s-1, and the uncertainty of background varies between 0.03 and 0.1 ppm (Table S1)." <<*
   *Comment: The uncertainty in the background stated here is much smaller than the range from 0.23 to 0.77 ppm. What is the reason for the differences?*

**Response:** The range from 0.23 to 0.77 ppm is for the 9 percentile bins (60th, 65th, 70th, 75th, 80th, 85th, 90th, 95th, 99th). The range from 0.03 and 0.1 ppm is for the selected percentile bins (75th, 80th, 85th, 90th) used to calculate uncertainty of background. More details are mentioned in the reply to comment 1.

5. *>> (5) The parameter A is one of the parameters to be fitted and not the line density. <<*
   *Comment: This is wrong. The fitting parameter "A" in your Eq. 3 is already the line density (in ppm/m), i.e. the area under the Gaussian curve.*
   *If I understand your approach correctly, you first fit Eq. (3) and then use the fitting parameters (k and b) to compute the XCO2 enhancement by subtracting the background. You then use Eq. (S2) to compute the line density from the XCO2 enhancement, which you convert to g/m using Eq. (2)? If this is correct, I wonder if you approach has any advantage and how your line density actually differs from using the fitting parameter "A" directly.*

**Response:** The line density unit is ppm*m. Thanks for the suggestion. We double check our line density calculation approach "the shaded area of Fig. R4" with the parameter A. We find the A is indeed same with the shaded area. Thus, we corrected the manuscript in L183-184: "A represents the line density which is same as the area under the fitted curve (Figure S1b) after removing the background."

| OCO2-case-id | Fit_A | the shaded area (ppm*m) |
|:---:|:---:|:---:|
| 1 | 9942.5 | 9942.5 |
| 2 | 47054.5 | 47054.5 |

| | | |
|---|---|---|
| 3 | 7751.5 | 7751.5 |
| 4 | 14192.9 | 14192.9 |
| 5 | 10006.3 | 10006.3 |
| 6 | 13667.2 | 13667.2 |
| 7 | 15027.8 | 15027.8 |
| 8 | 10039.1 | 10039.1 |
| 9 | 11277.3 | 11277.3 |
| 10 | 14055.9 | 14055.9 |
| 11 | 17315.4 | 17315.4 |
| 12 | 13880.5 | 13880.5 |
| 13 | 27847.6 | 27847.6 |
| 14 | 9745.0 | 9745.0 |
| 15 | 33348.9 | 33348.9 |
| 16 | 4205.0 | 4205.0 |
| 17 | 7477.3 | 7477.3 |
| 18 | 5960.1 | 5960.1 |
| 19 | 16962.3 | 16962.3 |
| 20 | 7140.0 | 7140.0 |
| 21 | 15855.3 | 15855.3 |
| 22 | 39681.9 | 39681.9 |
| 23 | 28281.4 | 28281.4 |
| 24 | 10867.8 | 10867.8 |
| 25 | 4991.5 | 4991.5 |
| 26 | 12369.0 | 12369.0 |
| 27 | 12085.9 | 12085.9 |
| 28 | 28443.2 | 28443.2 |
| 29 | 4301.3 | 4301.3 |
| 30 | 5222.4 | 5222.4 |

6.   *>> (9) We have added why WPBL provides better results in the revised manuscript L267-272: "[…]. The reason why the results using MERRA2 were worse (Fig. S4) is due to its low resolution, which cannot provide precise wind information for emission sources." <<*
*Comment: I think it would be important to add a reference to this statement.*

**Response:** We added the MERRA-2 dataset reference mentioning the resolution in L271-272: "The results using MERRA2 were worse (Fig. S4) due to its low resolution (GMAO, 2015), which cannot provide precise wind information for emission sources."

7.   *>> (16) We added explanations in the revised manuscript L378-382: "We found that the cross-sectional flux method has a larger variability than the GPM method. […] multiplied by the component of wind perpendicular to the orbit […]. <<*
*Comment: The explanation is still unclear to me. Do you mean that the limitation of the CFM is that you fit a symmetric Gaussian curve, but for a large angle between orbit and wind direction, the correct function would be an asymmetric Gaussian curve, which results in an additional source of uncertainty?*

**Response:** Yes. When the angle between the orbit and the wind direction is large, the actual shape is asymmetric Gaussian. However, the resolution of OCO observations is not sufficient to fully fit asymmetric Gaussian curves. In addition, the use of wind speed components perpendicular to the track also resulted in estimated emission errors. Thus, we corrected the manuscript in L334-335: "This is because when the angle between the orbit and the wind direction is large, the actual cross-section shape is asymmetric Gaussian. But the resolution of OCO observations is not sufficient to fully fit asymmetric Gaussian curves."